# Is This the Real Life, or Is This Just Laboratory? A Scoping Review of IMU-Based Running Gait Analysis

**DOI:** 10.3390/s22051722

**Published:** 2022-02-23

**Authors:** Lauren C. Benson, Anu M. Räisänen, Christian A. Clermont, Reed Ferber

**Affiliations:** 1Faculty of Kinesiology, University of Calgary, Calgary, AB T2N 1N4, Canada; araisanen@westernu.edu (A.M.R.); christian.clermont@ucalgary.ca (C.A.C.); rferber@ucalgary.ca (R.F.); 2Tonal Strength Institute, Tonal, San Francisco, CA 94107, USA; 3Department of Physical Therapy Education, College of Health Sciences—Northwest, Western University of Health Sciences, Lebanon, OR 97355, USA; 4Sport Product Testing, Canadian Sport Institute Calgary, Calgary, AB T3B 6B7, Canada; 5Cumming School of Medicine, Faculty of Nursing, University of Calgary, Calgary, AB T2N 1N4, Canada; 6Running Injury Clinic, Calgary, AB T2N 1N4, Canada

**Keywords:** biomechanics, wearable devices, injury, running conditions

## Abstract

Inertial measurement units (IMUs) can be used to monitor running biomechanics in real-world settings, but IMUs are often used within a laboratory. The purpose of this scoping review was to describe how IMUs are used to record running biomechanics in both laboratory and real-world conditions. We included peer-reviewed journal articles that used IMUs to assess gait quality during running. We extracted data on running conditions (indoor/outdoor, surface, speed, and distance), device type and location, metrics, participants, and purpose and study design. A total of 231 studies were included. Most (72%) studies were conducted indoors; and in 67% of all studies, the analyzed distance was only one step or stride or <200 m. The most common device type and location combination was a triaxial accelerometer on the shank (18% of device and location combinations). The most common analyzed metric was vertical/axial magnitude, which was reported in 64% of all studies. Most studies (56%) included recreational runners. For the past 20 years, studies using IMUs to record running biomechanics have mainly been conducted indoors, on a treadmill, at prescribed speeds, and over small distances. We suggest that future studies should move out of the lab to less controlled and more real-world environments.

## 1. Introduction

Wearable technology has been adopted among sports science researchers and practitioners to capture movement in the conditions in which sports take place [1]. Inertial measurement units (IMUs) are a type of wearable technology that can be used to measure running biomechanics [2]. The use of IMUs for real-world monitoring of running biomechanics may provide insights that are different from observations in controlled conditions [3,4,5]. Historically, the space and computational costs of onboard data storage and processing have created challenges for long-term monitoring of running biomechanics [2]. However, as device capabilities and approaches to big data have improved, the large amounts of data produced by IMUs have changed from a liability to an opportunity for real-world running biomechanical analyses.

While several editorials and commentaries have indicated the capability of IMUs to study running biomechanical gait patterns out of the laboratory and recommend that investigators do so [2,3,4,6,7], these suggestions were not based on systematic evidence. Therefore, we do not know how many studies are using IMUs to record running biomechanics and in which settings. In 2018, a systematic review identified only 14 studies that used wearables for running gait analysis for distances greater than 200 m [8]. As the use of wearables is a trending topic in the field of running biomechanics, we expect the number of studies that analyze running gait using IMUs in real-world settings to dramatically increase.

Thus, the purpose of this review is to systematically identify the scope of how IMUs are used to record running biomechanics in all settings. Our primary focus is on the conditions (i.e., location, surface, speed, and distance) in which IMUs capture running quality. Our secondary objectives were to identify the devices and sensors used, the calculated metrics and analyses from the IMU signals, the characteristics of the participants in the studies, and the study details such as the purpose and study design. By identifying the scope of IMU-based running biomechanical studies, we aim to mark the progress made and the steps that remain for analyzing running gait in real-world settings.

## 2. Materials and Methods

### 2.1. Registration

The review protocol was registered through the Open Science Framework on 24 August 2020 (https://osf.io/gsmvj/?view_only=cc97d0034c5341bca4ac181878770ec7, accessed on 16 February 2022).

### 2.2. Eligibility Criteria

This review was designed to capture all journal articles that used IMUs to assess gait quality during running, published in English since 2001. Exclusion criteria were: not original research article (e.g., review papers, conference proceedings, and dissertations); the study did not involve human subjects; the study did not involve running; running quality was analyzed as part of another athletic task (e.g., change of direction and playing a team sport); only spatiotemporal variables were analyzed (e.g., speed, cadence, and step length); there was no use of IMUs; the sole purpose of the study was the use of IMUs for any purpose other than gait analysis; the study primarily focused on development of new technology or methods rather than gait analysis; and running was only with the use of robotic orthoses, exoskeletons, or virtual reality environments.

### 2.3. Search Strategy

The search was executed in the scientific databases CINAHL, Embase, HealthSTAR, MEDLINE, PsycINFO, PubMed, SPORTDiscus and Web of Science. Databases were searched for articles related to IMUs and running using the following terms and logic: (Wearable Electronic Devices/OR Accelerometry/OR wearable* OR inertial sensor* OR inertial measurement unit* OR imu OR imus OR gyroscope* OR magnetometer* OR acceleromet*) AND (Running/OR running OR jogging), where/indicates a MESH term and * indicates the search term can have any ending. The final search was conducted on 24 April 2021.

### 2.4. Study Selection

One author (LCB) searched each database, combined the resulting records from each database, and performed initial screening for duplicates, format, and language. The records that passed initial screening were uploaded to an online review management platform (Covidence, Melbourne, Australia). Two authors (LCB and AMR) screened the title and abstract of all records for eligibility, with one author (CAC) serving as the tiebreaker. One author (LCB) obtained and uploaded the full text of the records that passed the title and abstract screening. The full-text review was conducted by two authors (LCB and AMR), and the reason for exclusion was indicated for articles deemed ineligible. In the case where multiple exclusion criteria were relevant, the criterion highest in the list above was chosen. One author (CAC) served as the tiebreaker for conflicts on whether an article should be included or excluded as well as conflicts on the selected reason for exclusion.

### 2.5. Data Extraction

Each included study was assigned to an author (LCB, AMR, and CAC) who extracted data related to the study, participants, conditions, device(s), and analysis.

#### 2.5.1. Conditions

##### Location

Categorized as: indoor, outdoor, or indoor and outdoor.

##### Running Surface

Categorized as: track, pavement or sidewalk, grass (includes real or artificial), trail (includes gravel), treadmill, floor or platform, or not controlled.

##### Speed

Categorized as: exact (minimum speed of 1.67 m/s except for incremental runs that started with walking but ended with running; recorded in units of m/s), relative—calculated (based on a race time or a specific test [e.g., VO2max and heart rate]), relative—subjective (self-selected or based on participant interpretation [e.g., maximal, slow, and moderate]), and not controlled (races or training runs).

##### Full Distance or Duration

For each surface, the complete distance or duration was calculated by multiplying the distance or duration by the number of trials and number of days and reported using units from the study. Exceptions for using the reported units for the full distance or duration include more than 180 s (converted to minutes) and more than 5000 m (converted to km).

##### Analyzed Distance

For each surface, the amount of IMU data analyzed was categorized as: single step or stride per trial for one or more trials, consecutive steps for less than 200 m per trial for one or more trials, consecutive steps over 200 m to 1000 m per trial for one or more trials, consecutive steps over more than 1000 m for one trial, consecutive steps over more than 1000 m for multiple trials. A trial was a repeated run on the same or different course, or a repeated or different segment run on the same or different days. If not provided, the analyzed distance was calculated from the reported speed. If only the number of steps or insufficient information was reported for determining the analyzed distance, equivalences between 200 m, 60 s, and 150 steps/min were used—200 m in 60 s corresponds with a speed of 3.33 m/s, which is a common intermediate running speed [9,10], and 150 steps/min is on the low end of preferred running cadence [11,12,13,14], representing a low threshold of number of steps that equates to 200 m.

#### 2.5.2. Device(s)

##### Brand and Model

As reported.

##### Device Location(s)

Categorized as: foot (any portion of foot or shoe), shank (includes tibia/shin, calf, ankle), thigh, lower back (includes pelvis, lumbar spine), upper back (includes thoracic, cervical spine), chest, arm (includes wrist), head. If multiple devices were placed on the same location (e.g., both feet), the location was only recorded once.

##### Sensors

The number of axes were recorded for each type of sensor: Accelerometer, gyroscope, magnetometer.

#### 2.5.3. Analysis

##### Statistical Approach

Categorized as: descriptive, inferential, or machine learning. Descriptive was only used when it was the only type of analysis performed. Machine learning was included only when it was used as part of the statistical approach and not to generate metrics (e.g., estimated ground reaction forces from accelerations using an artificial neural network).

##### Metrics

Categorized as: vertical/axial magnitude (e.g., peak and RMS), anterior–posterior magnitude (e.g., peak and RMS), medial–lateral magnitude (e.g., peak, RMS), resultant magnitude (e.g., peak and RMS), axis ratio (e.g., axis RMS/resultant RMS), variability—any axis (e.g., SD and CV), loading rate, power, PlayerLoad, shock attenuation—time domain, shock attenuation—frequency domain, frequency content, spectral power or spectral energy, stiffness, joint angles or ROM, joint angular velocity, segment rotation, segment rotation velocity, COM displacement (e.g., bounce, oscillation, and trajectory), COM change in velocity (e.g., braking), symmetry or regularity (based on autocorrelation of signal), stability (e.g., Lyapunov exponent), or entropy. Due to the large number of studies investigating shock absorption using an accelerometer placed on the tibia, when the axis for tibial acceleration magnitude was not specified, it was assumed to be vertical. For other situations where the axis of acceleration magnitude was not reported, it was assumed to be the resultant.

#### 2.5.4. Participants

##### Sex

Females, males, or females and males.

##### Type

Non-runner (includes sedentary, adults, recreational team sport athletes), recreational runner (described as a runner; includes runners with defined weekly mileage, well-trained runners), and competitive runner (competes at a high level; includes elite, member of a collegiate or higher sports team).

##### Injury Status

Injured or uninjured.

##### Age

The central tendency and variability of age were recorded across all participant types. If the study only reported age for each participant group, the overall mean age was calculated.

#### 2.5.5. Study Details

##### Country

Based on ethics approval, or if not reported, the first author’s first affiliation.

##### Study Design

Randomized controlled trial, quasi-experimental, case study, case series, case control, prospective cohort, retrospective cohort, or cross-sectional.

##### Purpose

Equipment intervention, training intervention, validity or reliability of metric(s), compare metrics, compare groups, identify changes due to fatigue, identify changes between sessions, identify changes between conditions, associate with injury or associate with performance. Some studies had multiple purposes.

### 2.6. Quality Assessment

A formal quality assessment was not part of this scoping review. However, we evaluated the amount of information reported (adequate or lacking) and the relevance to running and IMUs (appropriate or not appropriate) for each set of data extracted (study, participants, conditions, device(s), and analysis) of each study.

## 3. Results

### 3.1. Study Selection

A total of 16,023 records were identified across all databases (Figure 1) and 7336 records were excluded during title and abstract screening. Of the 402 full-text articles that were assessed, 171 were excluded and 231 studies met all eligibility criteria and were included.

### 3.2. Data Extraction

The complete data extracted for each included study are reported as an appendix. The conditions, devices, analysis, participants, and study details are summarized here with key details provided in Table 1, Table 2, Table 3, Table 4, Table 5 and Table 6.

#### 3.2.1. Conditions

Across the 231 included studies, running gait was analyzed in 286 different conditions; however, for 24 conditions, the analyzed distance, speed and/or location could not be classified due to lack of information. The 262 running conditions that could be classified consisted of 27 unique combinations of the specified categories for analyzed distance (one step or stride, <200 m, 200–1000 m, >1000 m single trial, >1000 m multiple trials), speed (exact, calculated, subjective, not controlled) and location (indoor, outdoor) (Figure 2).

The most common running condition was indoors at an exact speed with <200 m analyzed, accounting for 21% of all 262 conditions. Indoor running at a subjective speed with <200 m analyzed was the second most common condition at 20%. Indoor running at an exact or subjective speed with one step or stride analyzed accounted for 12% of all conditions. Overall, 72% of all conditions were indoors; and in 67% of all conditions, the analyzed distance was one step or stride or <200 m. Most of those studies were published between 2015 and 2021.

Studies with less controlled running conditions were primarily published after 2018. A total of 17% of all conditions included runs of >1000 m in a single or multiple trials, and speed was not controlled in races or training runs for 8% of all conditions. The least controlled condition—outdoor running with speed not controlled for multiple trials > 1000 m—accounted for 4% of all conditions.

The running conditions were grouped by surface and location (Figure 3). Overall, 49% of all conditions were indoors on a treadmill, and an additional 16% were indoors on a floor or platform. Outdoor pavement or sidewalk, trail, grass, and not controlled running surfaces combined for 19% of all conditions.

#### 3.2.2. Device(s)

There were 365 combinations of devices with specific sensor/axis composition and device locations on the body (Figure 4). The most common combination was a triaxial accelerometer on the shank (18% of all combinations) followed by a triaxial accelerometer on the lower back (13%). Across all sensors with any number of axes, the top three device locations were the shank, lower back and foot, accounting for 35%, 22% and 16% of all combinations, respectively. Across any number of axes, 70% of all combinations used an accelerometer only, and 22% of all combinations used all three sensors. A gyroscope was the only sensor for 1% of all combinations and was placed on the foot or shank.

Some studies used multiple devices, bringing the total number of devices to 251. Most devices (82%) were of research-grade. The remaining 18% of devices are commercially available and designed for public use and include adidas Run Genie, Catapult, DorsaVi, Garmin, Google Nexus, Lumo Run, Milestone Pod, Polar, RunScribe, Runteq Zoi, Stryd, and Zephyr BioHarness. These devices were commonly worn on the shoe or lower or upper back.

#### 3.2.3. Analysis

The reported metrics across all studies are shown in Figure 5. The most common metric was accelerometer magnitude, with vertical/axial, anterior–posterior, medial–lateral and resultant magnitude reported in 64%, 23%, 18% and 26% of all studies, respectively. (Note: studies often reported multiple metrics, and therefore the sum of the percentages is greater than 100%.) The acceleration quantity was also reported in metrics such as loading rate, PlayerLoad, power and stiffness, with loading rate being the most common (9% of all studies). Shock attenuation was reported in 7% of all studies using time domain calculations and in 8% of all studies using frequency domain calculations. Signal frequency content was reported in 7% of all studies and the spectral power or energy was reported in 9% of all studies. Signal consistency, represented by metrics such as variability, symmetry or regularity, entropy, and stability, was reported in 5% or less of all studies, each. Segment (including center of mass) or joint kinematics were reported in up to 12% of studies, with measures of displacement more common than measures of velocity.

In terms of a statistical approach, 91% of all studies used inferential statistics, 2% used machine learning, and 7% presented results descriptively. The most common metrics reported in studies that used a machine learning statistical approach were vertical/axial magnitude, anterior–posterior magnitude, and joint angles or range of motion.

#### 3.2.4. Participants

Half of the studies included male and female participants, 35% included males only, 3% included females only, and the sex of participants was not specified in 12% of all studies. Participants were uninjured in 99% of all studies. The mean participant age within a study ranged from 5 to 59 years, with an average of 27 years across all studies.

Recreational runners, non-runners and competitive runners were participants in 56%, 30% and 17% of all studies, respectively. (Note: some studies included multiple participant types, and therefore the sum of the percentages is greater than 100%.) The average number of participants reported for each participant type and sex is shown in Figure 6. With an average of 25 participants per study, recreational runners had the greatest number of participants per study, followed by non-runners (n = 20) then competitive runners (n = 14). This pattern was consistent when separated by males and females, and the average number of male recreational and competitive runners was greater than the average number of females in each group.

#### 3.2.5. Study Details

Studies were conducted in 27 different countries. The USA had the most studies (29%), followed by Canada at 9%. In total, 39% of studies were conducted in 11 European countries.

One-quarter of the studies had interventions: 24% were quasi-experimental and 1% were randomized controlled trials. Two-thirds of the interventions (17% of all studies) were equipment-based and one-third (9% of all studies) involved training interventions. The remaining 75% of studies were observational, and 95% of observational studies (71% of all studies) used a cross-sectional study design. Prospective and retrospective cohorts accounted for 2% and <1% of studies, respectively, and 2% of studies were case studies or case control. The most common purpose (22% of studies) was to determine the validity or reliability of metrics, and 16% of studies compared metrics. In 20% of studies, the purpose was to identify changes in conditions not related to fatigue or different sessions. Differences in gait due to group membership, fatigue and sessions were reported in 13%, 12% and 2% of studies, respectively. Associations of gait metrics with performance and injury were reported for less than 2% of studies, each. (Note: some studies had multiple purposes, and therefore the sum of the percentages is greater than 100%).

### 3.3. Quality Assessment

There was an adequate amount of information provided for the conditions (92% of studies), devices (91%), analysis (93%), participants (83%) and study details (100%). Additionally, the relevance to running and IMUs was deemed appropriate for the conditions (99% of studies), devices (100%), analysis (100%), participants (98%) and study details (99%).

## 4. Discussion

The primary purpose of this review was to systematically identify how IMUs are used to record running biomechanics across real-world settings and describe the conditions in which IMU data were collected. Identifying the characteristics of IMU-based running biomechanical studies serves to mark the progress made and the steps that remain for analyzing running gait in real-world settings.

### 4.1. Running Environments

Laboratory-based conditions are controlled and are often different from typical running conditions, as most runners complete their runs outdoors [243]. Additionally, loads vary with each stride and a runner’s load capacity changes throughout a running session [244], suggesting that assigning the same estimated load to each stride is not a suitable approximation for the cumulative load in a running session. Therefore, it is important to monitor running in actual real-world conditions, including over long distances. Yet, despite the portability of IMUs [6,7], one of the main findings of this review is that running biomechanics are mainly recorded with IMUs indoors, on a treadmill, at prescribed speeds, and over small distances. Furthermore, the majority of studies that investigated running in artificial environments have been published recently; there has not been a trend away from laboratory-based conditions over time. It is unclear why researchers are using IMUs to record running, but still have participants running in the laboratory, at controlled speeds, on treadmills and/or over short distances. If the purpose of these devices is to capture real-world running, we suggest that the research in this area should move out of the lab to less controlled environments.

Several of the included studies compared running quality between surfaces, and the findings underscore the need to observe runners in their actual running environment. More unstable surfaces lead to less regularity and greater variability during running [5,142], and the variance in outdoor data cannot be explained by indoor measures [31,95]. Moreover, it is likely that not all metrics differ between the running conditions [245]. For example, there was no difference in running power on a track compared to a treadmill [166]. Among the four studies that compared tibial acceleration between treadmill and outdoor running, the acceleration magnitude was either lower [241], greater [31,95], or not different [84,241] in outdoor conditions compared to on the treadmill, but in only one case did the outdoor conditions represent an uncontrolled running environment [95]. We suggest that rather than estimating what it is like to run outdoors, it would be helpful to use IMUs during actual training runs, over longer distances and on surfaces that represent real-world running.

To our point, starting from 2015, some studies have followed athletes for uncontrolled training runs or races [188,200,201,202,206,207,210,211,212,213,220,221,225,226,227,230]. A myriad of external factors, such as weather, traffic, and surface conditions, could influence how someone runs and therefore, it is crucial to capture running patterns in the same settings that runners *actually* run. Additionally, just as multiple trials are often used in a laboratory setting, multiple runs are needed to establish running patterns in uncontrolled settings [213].

### 4.2. IMU Considerations

The ability to collect accurate and useful metrics from IMUs depends on the desired metrics, the sensor specifications, device placement, running styles, and user capabilities [4]. IMUs intended for long term monitoring need to be user-friendly. The commercial devices in the included studies were worn on the foot or upper or lower back. In contrast, the most common position for devices among all included studies was on the shank, where tibial acceleration in one or multiple axes was recorded. Tibial accelerations have been used in the context of stress fractures as well as to gauge impact forces at the shank and how they are distributed along the kinetic chain [32]. While devices designed for consumer use have not been developed for placement on the shank, a research-grade device was used to record tibial accelerations of nearly 200 runners during a marathon [95]. Future investigations of impact forces in actual running conditions should consider devices and placement that can be easily applied during long-term monitoring.

The metrics reported from accelerometer sensors, such as the magnitude of acceleration, loading rate and shock attenuation, are similar to metrics obtained from force plates. When the gyroscope and/or magnetometer sensors in an IMU are used, the reported metrics provide information on the kinematics, including segment and joint rotations [28,48,51,53,55,76,102,106,113,114,115,127,128,129,134,140,144,145,148,157,181,183,186,188,190,192,200,201,210,211,212,213,216,217,219,221,228,235]. While it is typical for IMUs to contain multiple sensors, most included studies only used an accelerometer sensor, limiting the reported metrics to those that resemble force plate metrics.

Many of the included studies were conducted in indoor settings because the purpose of the study was to evaluate the validity and reliability of IMU-based metrics compared to metrics from force plates or motion capture systems. Assessing strength of the validity and reliability was not within the scope of this review; however, devices that demonstrate adequate validity and reliability can be used in the field. Additionally, while metrics reported from an IMU are often chosen to be similar to metrics from force plates and motion capture systems, it is possible to report metrics specific to IMU signals (e.g., entropy, regularity, and symmetry) that monitor movement quality [246].

### 4.3. Changing Running Biomechanics

It is expected that equipment or training interventions that lead to changes in running biomechanics are needed to change injury rates. However, there is limited or conflicting evidence on the relationship between modifications of running biomechanics and running injuries [6,72,73]. It is also possible that lack of clarity on running injury risk factors is related to evaluation of biomechanical metrics in a laboratory setting before and after an intervention or injury observation, and not in the runner’s natural environment [2].

Short-term changes, observed within a laboratory session, may show how training or equipment interventions can change running patterns [72,151]. Some studies use an intervention that is more long term to allow for adaptation and assess movement patterns at baseline and follow up to observe changes [20,154]. If biomechanics are only recorded in a single session, or at baseline and follow up, it is possible to use laboratory equipment (e.g., force plates and motion capture), but this does not reflect how runners run during real-world conditions. The benefit of IMUs is that movement patterns can be measured during the intervention period in actual running settings to monitor changes over time. Yet just two of the intervention studies from this review analyzed metrics from IMUs during an intervention (i.e., not just pre- and post-intervention) that was greater than one session, plus the intervention runs were conducted on a treadmill in both studies [154,218].

IMUs can also be used to observe changes in running patterns throughout a single run. In studies investigating changes in running biomechanics due to fatigue, it is common to have participants run to the point of exhaustion. Reaching a state of exhaustion as defined in a study may occur in some training runs or races, but it is likely not a typical running strategy for all runners. Thus, it is important to look at how running patterns change during actual training runs. More prospective or retrospective studies are also needed that look at how running patterns change over time, especially when those changes precede an injury [4,8]. Only five of the included studies included injured runners [24,144,152,208,221]. Due to pain, running in an injured state is likely not representative of running prior to injury. While some included studies involved runners that were previously injured and others looked at runners that were eventually injured, the data on the running patterns were only observed at a point when the runners were not injured. Regardless, IMUs can facilitate continuous monitoring that will allow for observation of changes in running patterns that lead to injury.

### 4.4. Participant Characteristics

Over 75% of runners use wearables, and most runners use wearable technology to monitor spatiotemporal parameters, such as distance or speed [247,248,249]. Competitive runners are more likely to use wearables to monitor running form or biomechanics than recreational runners [249]. Even if runners are not personally using IMUs that monitor their biomechanics, based on the results of survey studies, runners have a large appetite for using and consuming data from wearable technology [247,249]. Yet the number of participants in the included studies is low. Considering the popularity of running and runners’ attitudes towards wearable technology, investigations of real-world running biomechanics should be able to recruit large numbers of participants. A bigger pool of participants will enable better comparisons across participant types and consider sex differences as injury rates differ between sexes [250]. Based on race participation statistics, there are more female than male runners [251]. However, consistent with previous findings that show females are underrepresented in sport and exercise medicine research [252], we found that the running and IMU literature is also heavily focused on male runners, with only 3% of studies being female specific.

### 4.5. Limitations

There are some limitations to this review, based on the exclusion criteria. First, the only type of wearable technology considered was IMUs. Limiting the search to only IMUs excluded studies that only utilized GPS devices, which are very common among runners [249]. Additional types of wearable technology that were not included in this review are heart rate monitors, mobile phone apps that did not utilize the phone’s IMU sensors, and pressure-sensing insoles. Second, studies were excluded if they only reported spatiotemporal metrics. IMUs can be used to derive valid and reliable spatiotemporal stride parameters that capture running quantity [253]; however, load magnitude and distribution are also needed on a per stride basis to evaluate injury risk [244]. Finally, we excluded studies that focused on the development of new technology or methods, which eliminated some studies that reported novel machine learning algorithms. Whilst wearable technology is a growing field, future advancements will hopefully improve our ability to monitor real-world running.

There was no meta-analysis or formal quality assessment of each study as these are not expected for a scoping review. Based on our subjective evaluation, nearly all studies were appropriate to the topic of running and IMUs and contained adequate information for inclusion in this scoping review. Most likely, a more rigorous evaluation of study quality would have revealed overall weak levels of evidence across this field of study. We leave it to future systematic reviews and meta-analyses of specific outcomes and populations to use objective protocols for evaluating study quality.

## 5. Conclusions

Despite the portability of IMUs, one of the main findings of this review is that running biomechanics are mainly recorded with IMUs indoors, on a treadmill, at prescribed speeds, and over small distances. While it is challenging to collect data in real-world conditions due to the myriad of extrinsic factors such as weather, traffic, and surface conditions, our results indicate the vast majority of studies do not capture running biomechanical data in the same settings that runners actually run. Moreover, while it is typical for IMUs to contain multiple sensors, most included studies only used data derived from the accelerometer sensor and most studies involved placement of the IMU at the shank. Finally, the number of participants in the included studies is low and our findings show that research is still heavily focused on male runners, with only 3% of studies being female specific. Overall, considering that the purpose of IMU devices is to capture real-world running, we suggest that future research in this area should move out of the lab to less controlled and more real-world environments.

## Figures and Tables

**Figure 1 sensors-22-01722-f001:**
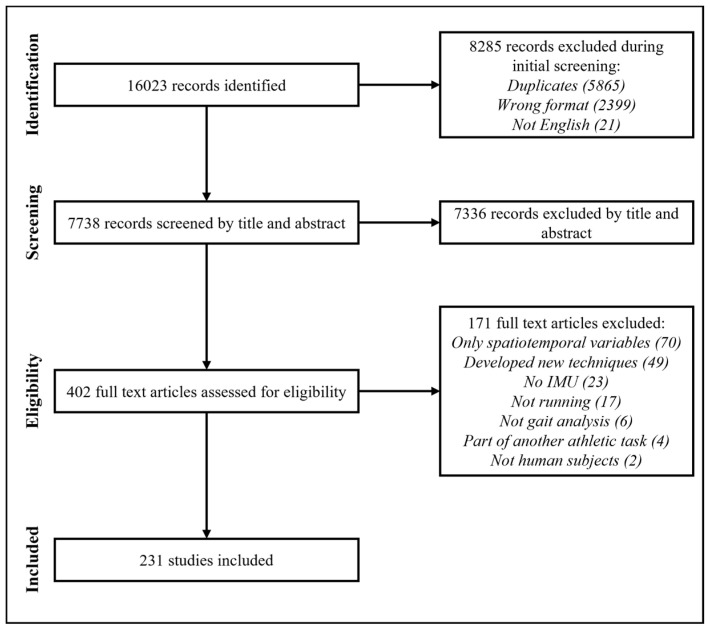
Flowchart of the study selection process.

**Figure 2 sensors-22-01722-f002:**
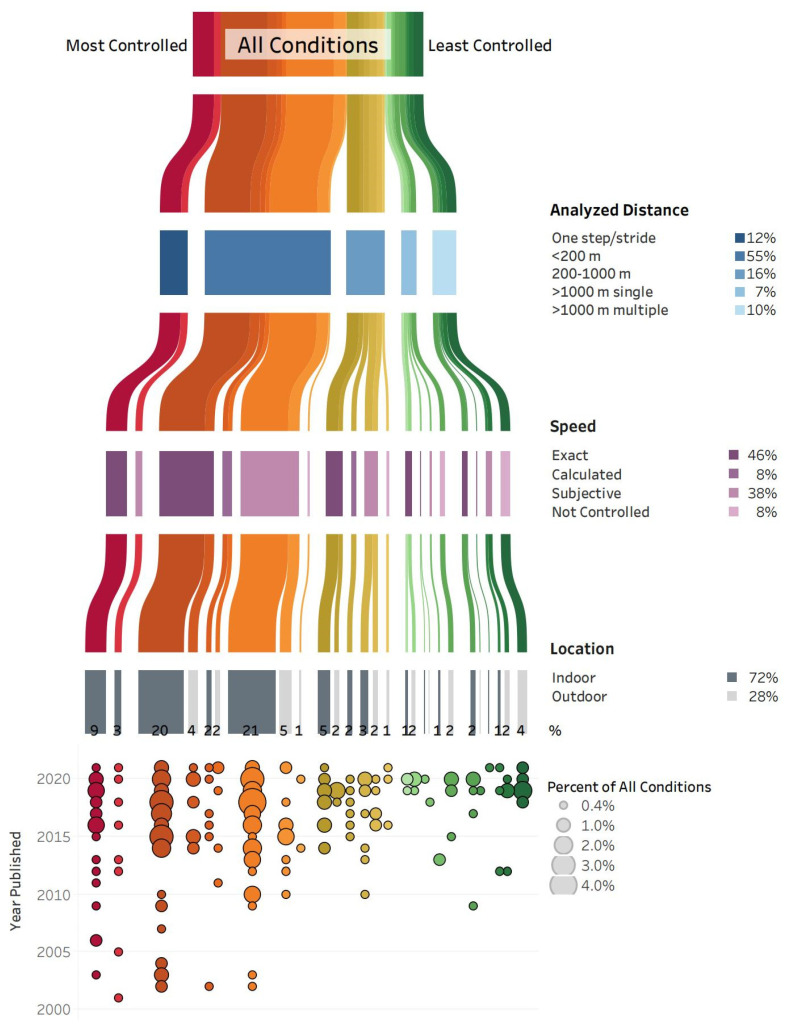
All conditions (262 across 231 studies) are grouped according to the analyzed distance, speed, and location. The condition groups are ranked from most controlled (red, on the left) to least controlled (green, on the right), and the width of each line corresponds to the percent of all conditions within that group. The degree of control is based on the analyzed distance (one step/stride is more controlled than >1000 m), speed (exact is more controlled than not controlled), and location (indoor is more controlled than outdoor). The percent of all conditions for each category of analyzed distance (shades of blue), speed (shades of purple), and location (shades of grey) is reported. In the bottom panel, the percent of all conditions within each group are further separated by year the study was published, with larger circles corresponding to a greater percent of all conditions. Note: 24 conditions that could not be categorized due to lack of information are not included in this figure.

**Figure 3 sensors-22-01722-f003:**
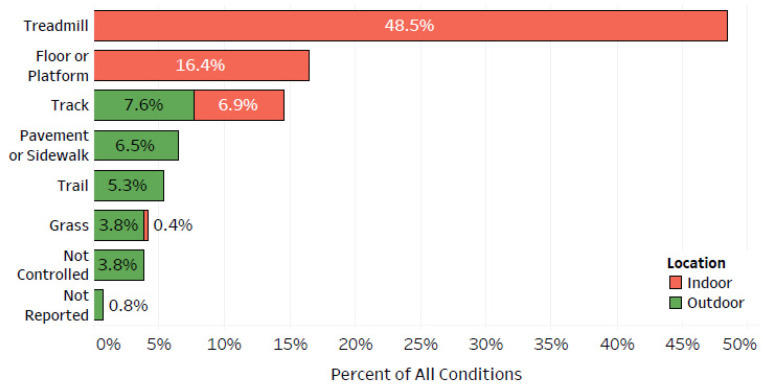
The percent of all conditions (262 across 231 studies) by running surface and location (indoor, outdoor).

**Figure 4 sensors-22-01722-f004:**
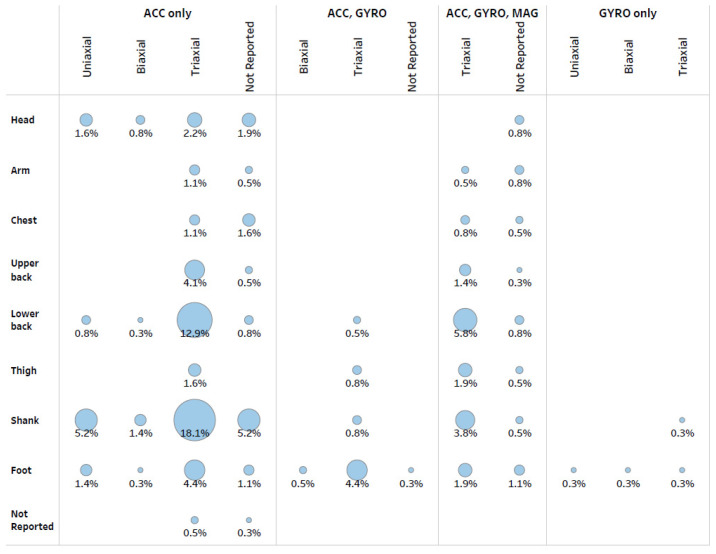
The percent of all combinations of devices and locations of devices on the body (365 combinations across 231 studies) by body location and sensor/axis composition. The circles are sized relative to percentage to provide visual comparisons for the frequency of device and location combinations. ACC = accelerometer, GYRO = gyroscope, and MAG = magnetometer.

**Figure 5 sensors-22-01722-f005:**
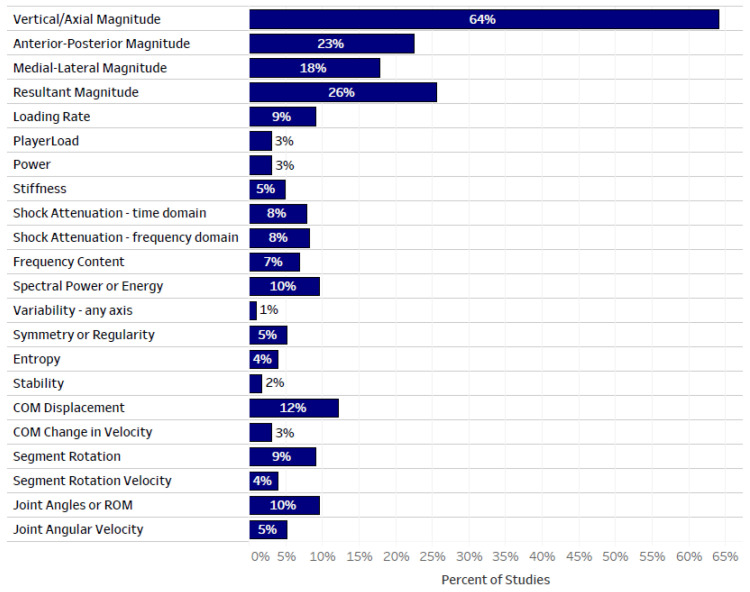
The percent of all studies that reported each metric. Note: studies often reported multiple metrics, and therefore the sum of all percentages is greater than 100%. COM = center of mass; ROM = range of motion.

**Figure 6 sensors-22-01722-f006:**
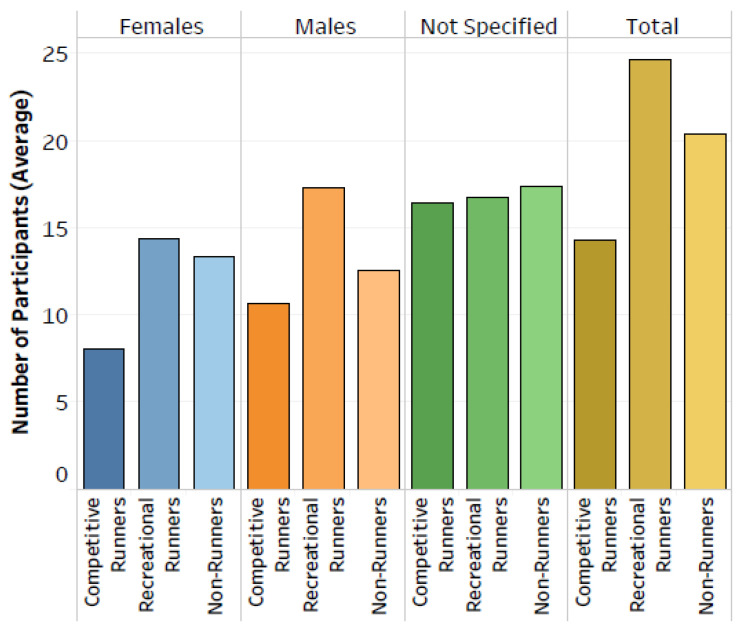
The average number of participants for each participant type and sex. Averages are reported across studies that included the given participant type and sex.

**Table 1 sensors-22-01722-t001:** Study characteristics where the analyzed distance is one step/stride.

Ref.	Author	Year	Location	Surface	Speed	Distance/Duration	Type	Number (Sex)	Overall Age	Metric(s)
[15]	Aubol, et al.	2020	Indoor	Floor	3.0 m/s	30 m	Rec.	19 (10 F, 9 M)	Mean: 31; SD: 6	VT, Res.
[16]	Blackah, et al.	2013	Indoor	Treadmill	3.83 m/s	2 min	Rec.	9 (0 F, 9 M)	Mean: 19; SD: 1	VT, Shock-time, Shock-frequency
[17]	Boyer and Nigg	2006	Indoor	Floor	3 m/s	320 m	Non	5 (0 F, 5 M)	Mean: 24.6; SD: 2.5	VT, Shock-frequency
[18]	Chadefaux, et al.	2019	Indoor	Floor	3.1 m/s	175 m	Rec.	10 (0 F, 10 M)	Mean: 21; SD: 3	VT, AP, ML, Res., Freq.
[19]	Clansey, et al.	2012	Indoor	Floor	4.5 m/s	270 m	Rec.	21 (0 F, 21 M)	Mean: 36.2; SD: 12.5	VT, Loading Rate
[20]	Crowell and Davis	2011	Indoor	Floor	3.7 m/s	230 m	Rec.	10 (6 F, 4 M)	Mean: 26; SD: 7	VT, Loading Rate
[21]	Edwards, et al.	2019	Indoor	Floor	3.3., 5.0, and 6.7 m/s	≥450 m	Comp.	10 (0 F, 10 M)	Mean: 21; SD: 2	VT
[22]	Gil-Rey, et al.	2021	Indoor	Track	incremental from 1.69 m/s	≥40 min	Non	82 (82 F, 0 M)	Mean: 59.1; SD: 5.4	VT, AP, ML
[23]	Hagen, et al.	2009	Indoor	Floor	3.3 m/s	NR	Rec.	20 (0 F, 20 M)	Mean: 32; SD: 10	VT
[24]	Havens, et al.	2018	Indoor	Floor	self-selected	90 m	Non	14 (7 F, 7 M)	Mean: 29; SD: 12	VT
[25]	Higgins, et al.	2021	Indoor	Floor	self-selected	368 m	Non	30 (15 F, 15 M)	Mean: 23.0; SD: 4.5	VT, Res.
[26]	Lam, et al.	2018	Indoor	Floor	3.0 and 6.0 m/s	230 m	Comp.	18 (0 F, 18 M)	Mean: 25.0; SD: 2.3	VT
[27]	Laughton, et al.	2003	Indoor	Floor	3.7 m/s	≥57 m	Rec.	15 (NS)	Mean: 22.46; SD: 4	VT
[28]	Mavor, et al.	2020	Indoor	Floor	self-selected	30 m	Non	18 (9 F, 9 M)	Mean: 23.7; SD: 3.44	Joint ROM
[29]	Meinert, et al.	2016	Indoor	Floor	2.9 m/s	30 m	Rec.	20 (0 F, 20 M)	Mean: 22.7; SD: 2.9	VT, Res., Shock-frequency, Spectral Energy
			Indoor	Treadmill	2.9 m/s	10 s				
[30]	Mercer, et al.	2005	Indoor	Floor	comfortable, faster, slower	800 m	Non	6 (NS)	Mean: 26; SD: 4.0	VT, Shock-time
[31]	Milner, et al.	2020	Indoor	Floor	3.0 m/s	≥30 m	Rec.	19 (10 F, 9 M)	Mean: 31; SD: 6	VT, AP, ML, Res., Seg. Rot.
[32]	Milner, et al.	2006	Indoor	Floor	3.7 m/s	115 m	Rec.	40 (40 F, 0 M)	Mean: 26; SD: 9	VT
[33]	Nedergaard, et al.	2018	Indoor	Floor	2, 3, 4, and 5 m/s	≥64 m	Non	20 (0 F, 20 M)	Mean: 22; SD: 4	Res.
[34]	Ogon, et al.	2001	Indoor	Floor	slow	144 m	Non	12 (5 F, 7 M)	Mean: 32.9; SD: 7.9	Loading Rate
[35]	Rowlands, et al.	2012	Indoor	Floor	self-selected	320 m	Non	10 (5 F, 5 M)	Mean: 29.4; SD: 7.3	VT, Res., Loading Rate
[36]	Sayer, et al.	2020	Indoor	Floor	2.8–3.2 m/s	NR	Non	64 (64 F, 0 M)	Mean: 13.7; SD: 2.3	VT, AP
[37]	Sinclair and Dillon	2016	Indoor	Floor	4 m/s	110 m	Rec.	12 (0 F, 12 M)	Mean: 23.59; SD: 2	VT, Loading Rate
[38]	Sinclair and Sant	2017	Indoor	Floor	4 m/s	NR	Non	13 (0 F, 13 M)	Mean: 27.81; SD: 7.02	VT, Loading Rate
[39]	Sinclair, et al.	2016	Indoor	Floor	4 m/s	NR	Rec.	12 (0 F, 12 M)	Mean: 23.11; SD: 5.01	VT, Loading Rate
[40]	Sinclair, et al.	2015	Indoor	Floor	4 m/s	110 m	Rec.	12 (12 F, 0 M)	Mean: 21.45; SD: 2.98	VT, Loading Rate
[41]	Sinclair, et al.	2017	Indoor	Grass	4 m/s	NR	Comp.	12 (0 F, 12 M)	Mean: 22.47; SD: 1.13	VT
[42]	Thompson, et al.	2016	Indoor	Floor	self-selected	450 m	Rec.	10 (5 F, 5 M)	Mean: 26; SD: 7.3	Res.
[43]	Trama, et al.	2019	Indoor	Track	2.22, 2.92, 3.61, and 4.31 m/s	≥60 m	Rec.	20 (0 F, 20 M)	Mean: 23.9; SD: 2.1	VT, Shock-frequency, Freq., Spectral Energy
[44]	Van den Berghe, et al.	2019	Indoor	Floor	2.55, 3.20, and 5.10 m/s	768 m	Rec.	13 (NS)	NR	VT, Res.
[45]	Wundersitz, et al.	2013	Indoor	Floor	maximal	50 m	Non	17 (5 F, 12 M)	Mean: 21; SD: 2	VT, Res.

Note: NR = not reported; Pavement = pavement or sidewalk; Floor = floor or platform; Rec. = recreational; Comp. = competitive; Non = non-runners; Disp. = displacement; Δv = change in velocity; Sym. or Reg. = symmetry or regularity; Res. = resultant magnitude; VT = vertical/axial magnitude; AP = anterior–posterior magnitude; ML = medial–lateral magnitude; Seg. Rot. = segment rotation; Shock—time = shock attenuation—time domain; Shock—frequency = shock attenuation—frequency domain; Joint ROM = joint angles or range of motion; Joint ω = joint angular velocity; Freq. = frequency content.

**Table 2 sensors-22-01722-t002:** Study characteristics where the analyzed distance is <200 m.

Ref.	Author	Year	Location	Surface	Speed	Distance/Duration	Type	Number (Sex)	Overall Age	Metric(s)
[46]	Adams, et al.	2016	Indoor	Treadmill	comfortable	2 min	Rec.	20 (8 F, 12 M)	Mean: 30.0; SD: 7.0	COM Disp.
[47]	Adams, et al.	2018	Indoor	Treadmill	comfortable	90 s	Rec.	20 (NS)	NR	COM Disp.
[48]	Argunsah Bayram, et al.	2021	Indoor	Treadmill	preferred	15 min	Non	24 (10 F, 14 M)	Mean: 22.2; SD: 0.9	Joint ROM, COM Disp., Sym. or Reg.
[49]	Armitage, et al.	2021	Indoor	Floor	sprint	21 m	Non	16 (0 F, 16 M)	Mean: 17; SD: 1	Res.
			Indoor	Treadmill	3.0 m/s	1 min				
[50]	Backes, et al.	2020	Indoor	Treadmill	2.78 and 3.33 m/s	2 min	Rec.	39 (6 F, 33 M)	Mean: 41.8; SD: 9.8	COM Disp.
[51]	Bailey and Harle	2015	Indoor	Treadmill	2.3, 2.7, 3.0, and 3.4 m/s	6 min	Rec.	3 (1 F, 2 M)	NR	Res., Seg. Rot.
			Outdoor	Grass	steady state	100 m				
			Outdoor	Pavement	steady state	100 m				
			Outdoor	Track	steady state	100 m				
			Outdoor	Trail	steady state	100 m				
[52]	Barnes, et al.	2021	Outdoor	Grass	2.1, 2.9, and 4.4 m/s	140 m	Comp.	29 (0 F, 29 M)	Mean: 25.2; SD: 3.5	PlayerLoad
[53]	Bastiaansen, et al.	2020	NR	NR	maximal sprint	90 m	Non	5 (0 F, 5 M)	Mean: 22.5; SD: 2.1	Joint ROM, Joint ω
[54]	Benson, et al.	2018	Indoor	Treadmill	preferred	2 min	Rec.	44 (18 F, 26 M)	Mean: 13.9; SD: 12.3	VT, AP, ML, Res.
[55]	Bergamini, et al.	2012	Outdoor	Track	sprint	180 m	Comp.	5 (2 F, 3 M)	NR	Res., Joint ω
							Rec.	6 (2 F, 4 M)		
[56]	Boey, et al.	2017	NR	Pavement	3.06 m/s	90 m	Rec.	23 (11 F, 12 M)	Mean: 23.3; SD: 3.0	VT
			NR	Track	3.06 m/s	90 m	Non	12 (6 F, 6 M)		
			NR	Trail	3.06 m/s	90 m				
[57]	Boyer and Nigg	2007	Indoor	Track	4.8 m/s	NR	Non	13 (NS)	NR	VT, Freq.
[58]	Boyer and Nigg	2004	Indoor	Floor	2.0, 3.0, 4.0, and 5.5 m/s	960 m	Non	10 (0 F, 10 M)	Mean: 25; SE: 4.2	VT, Freq.
[59]	Brayne, et al.	2018	Indoor	Treadmill	2.5, 3.5, and 4.5 m/s	120 s	Rec.	13 (0 F, 13 M)	Mean: 30; SD: 7	VT
[60]	Buchheit, et al.	2015	Indoor	Treadmill	2.78, 4.72, and 6.67 m/s	450 s	Non	1 (0 F, 1 M)	Exact: 36; NA	Stiffness
[61]	Butler, et al.	2003	Indoor	Floor	3.4 m/s	375 m	Rec.	15 (NS)	NR; Range: 18–45	VT
[62]	Camelio, et al.	2020	Indoor	Treadmill	preferred	36 min	Rec.	17 (9 F, 8 M)	Mean: 27; SD: 7	VT
[63]	Carrier, et al.	2020	Indoor	Treadmill	self-selected	6 min	Rec.	17 (8 F, 9 M)	Mean: 28.1; SD: 7.38	COM Disp.
[64]	Castillo and Lieberman	2018	Indoor	Treadmill	3.0 m/s	2 min	Non	27 (13 F, 14 M)	NR; Range: 18–45	Shock-frequency, Spectral Energy, Joint ROM, Joint ω
[65]	Chen, et al.	2021	Indoor	Treadmill	2.5 m/s	3 min	Non	24 (0 F, 24 M)	NR	Spectral Energy
[66]	Cheung, et al.	2018	Indoor	Treadmill	typical	12 min	Rec.	16 (5 F, 11 M)	Mean: 28.3; SD: 6.2	VT, Loading Rate
[67]	Cheung, et al.	2019	Indoor	Treadmill	2.78 m/s	10 min	Rec.	14 (7 F, 7 M)	Mean: 26.4; SD: 11.2	VT, Loading Rate
[68]	Ching, et al.	2018	Indoor	Treadmill	self-selected	20 min	Rec.	16 (9 F, 7 M)	Mean: 25.1; SD: 7.9	VT, Loading Rate
[69]	Chu and Caldwell	2004	Indoor	Treadmill	4.17 m/s	≥75 s	Rec.	10 (0 F, 10 M)	Mean: 26; SD: 6	VT, Shock-frequency, Spectral Energy
[70]	Clark, et al.	2010	Indoor	Treadmill	2.78 m/s	12 min	Non	36 (36 F, 0 M)	Mean: 30.3; SD: 5.8	VT, AP, ML
[71]	Creaby and Frattenovich Smith	2016	Indoor	Treadmill	3 m/s	50 min	Rec.	22 (0 F, 22 M)	Mean: 25.4; SD: 6.2	VT
[72]	Crowell, et al.	2010	Indoor	Treadmill	self-selected	25 min	Rec.	5 (5 F, 0 M)	Mean: 26; SD: 2	VT, Loading Rate
[73]	Day, et al.	2021	Indoor	Treadmill	3.8, 4.1, 4.9, and 5.4 m/s	NR	Comp.	30 (21 F, 9 M)	NR	VT, Spectral Energy
[74]	De la Fuente, et al.	2019	Indoor	Treadmill	typical	10 min	Rec.	20 (0 F, 20 M)	Mean: 30.5; SD: 9.3	VT, Res., Freq.
[75]	Deflandre, et al.	2018	Indoor	Treadmill	2.22 and 4.44 m/s	6 min	Rec.	20 (0 F, 20 M)	Mean: 26; SD: 9.5	Stiffness, Joint ROM, COM Disp., Sym. or Reg.
			Outdoor	Grass	2.22, 2.78, and 3.33 m/s	720 m	Non	10 (0 F, 10 M)		
[76]	DeJong and Hertel	2020	Indoor	Treadmill	2.68 and 3.6 m/s	180 s	Comp.	20 (12 F, 8 M)	Mean: 20; SD: 2	Joint ROM
[77]	Derrick, et al.	2002	Indoor	Treadmill	3.2 km pace	run to exhaustion	Rec.	10 (NS)	Mean: 25.8; SD: 7.0	VT, Shock-time, Shock-frequency, Spectral Energy, Joint ω, Seg. Rot.
[78]	Dufek, et al.	2009	Indoor	Treadmill	preferred, 10% slower	102 s	Rec.	14 (7 F, 7 M)	Mean: 24.9; SD: 4	VT, Shock-time
[79]	Eggers, et al.	2018	Indoor	Track	3.33 m/s	400 m	Non	17 (7 F, 10 M)	NR; Range: 18–40	VT, Stiffness, COM Disp.
[80]	Encarnación-Martínez, et al.	2020	Indoor	Treadmill	3.89 m/s	4 min	Rec.	17 (0 F, 17 M)	Mean: 28.7; SD: 8.3	VT, Shock-frequency, Spectral Energy
[81]	Encarnación-Martínez, et al.	2018	Outdoor	Grass	3.33 and 4.00 m/s	480 m	Non	12 (0 F, 12 M)	Mean: 24.3; SD: 3.7	VT, Shock-time
[82]	Encarnación-Martínez, et al.	2021	Indoor	Treadmill	2.78 m/s	10 min	Non	30 (10 F, 20 M)	Mean: 26.3; SD: 7.0	VT
[83]	Friesenbichler, et al.	2011	Outdoor	NR	10 km pace	run to exhaustion	Rec.	10 (7 F, 3 M)	Mean: 31.7; SD: 7.3	VT, AP, ML, Freq., Spectral Energy
[84]	Fu, et al.	2015	Indoor	Treadmill	3.33 m/s	6 min	Rec.	13 (0 F, 13 M)	Mean: 23.7; SD: 1.2	VT
			Outdoor	Grass	3.33 m/s	90 m				
			Outdoor	Pavement	3.33 m/s	90 m				
			Outdoor	Track	3.33 m/s	90 m				
[85]	Gantz and Derrick	2018	Indoor	Treadmill	self-selected	≥6 min	Rec.	16 (7 F, 9 M)	Mean: 22.9; SD: 3.3	VT, Shock-frequency, Spectral Energy
[86]	Garcia, et al.	2021	Outdoor	Pavement	self-selected	200 m	Rec.	15 (12 F, 3 M)	Mean: 27.7; SD: 9.1	VT, AP, ML, Res., Shock-time
			Outdoor	Trail	self-selected	400 m				
[87]	García-Pérez, et al.	2014	Indoor	Treadmill	4 m/s	800 m	Rec.	20 (9 F, 11 M)	Mean: 34; SD: 8	VT, Loading Rate, Shock-time
			Outdoor	Track	4 m/s	800 m				
[88]	Giandolini, et al.	2014	Indoor	Treadmill	3.89 and 4.44 m/s	8 min	Rec.	48 (18 F, 30 M)	Mean: 38.4; SD: 6.7	VT
			Outdoor	Trail	2.78 and 3.33 m/s	6 min				
[89]	Giandolini, et al.	2013	Indoor	Treadmill	self-selected	60 min	Non	30 (8 F, 22 M)	Mean: 18.3; SD: 4.5	VT
[90]	Glassbrook, et al.	2020	Indoor	Treadmill	60%–100% of maximal	435 s	Non	16 (6 F, 10 M)	Mean: 24.5; SD: 4.5	Res.
[91]	Gullstrand, et al.	2009	Indoor	Treadmill	2.78, 3.33, 3.89, 4.44, 5.00, 5.56, and 6.11 m/s	≥210 s	Rec.	13 (0 F, 13 M)	Mean: 22.7; NR	COM Disp.
[92]	Hardin and Hamill	2002	Indoor	Treadmill	3.4 m/s	30 min	Rec.	24 (0 F, 24 M)	NR	VT
[93]	Iosa, et al.	2014	Indoor	Floor	self-selected	10 m	Non	25 (NS)	Mean: 15.3; SD: 3.9	VT, AP, ML, Stability
[94]	Iosa, et al.	2013	Indoor	Floor	self-selected	10 m	Non	40 (16 F, 24 M)	Mean: 5.5; SD: 2.5	VT, AP, ML, Res., Freq.
[95]	Johnson, et al.	2020	Indoor	Treadmill	90% marathon pace	30 s	Rec.	192 (87 F, 105 M)	Mean: 44.9; SD: 10.8	VT, Res.
			Outdoor	Pavement	not controlled	42.2 km				
[96]	Johnson, et al.	2021	Indoor	Treadmill	self-selected	16 s	Rec.	18 (8 F, 10 M)	Mean: 33; SD: 11	AP, ML
[97]	Johnson, et al.	2020	Indoor	Treadmill	self-selected	32 s	Rec.	18 (8 F, 10 M)	Mean: 33; SD: 11	VT, Res., Loading Rate
[98]	Kawabata, et al.	2013	Indoor	Treadmill	slow, preferred, fast	18 min	Non	13 (0 F, 13 M)	Mean: 23.3; SD: 0.6	VT, AP, ML
			Outdoor	Track	slow, preferred, fast	4800 m				
[99]	Kenneally-Dabrowski, et al.	2018	Indoor	Track	maximal sprint	120 m	Comp.	13 (0 F, 13 M)	Mean: 23.8; SD: 2.4	VT
[100]	Khassetarash, et al.	2015	Indoor	Treadmill	4 m/s	run to exhaustion (10 km max)	Comp.	8 (0 F, 8 M)	Mean: 26; SD: 3.6	Shock-frequency
[101]	Kobsar, et al.	2014	Indoor	Treadmill	self-selected	135 s	Rec.	42 (42 F, 0 M)	Mean: 33; SD: 6	VT, AP, ML, Res., Freq., Axis Ratio, Sym. or Reg.
[102]	Koldenhoven and Hertel	2018	Indoor	Treadmill	comfortable	1.5 miles	Rec.	12 (6 F, 6 M)	Mean: 23.1; SD: 5.5	Seg. Rot., Seg. Rot. Velocity
[103]	Le Bris, et al.	2006	NR	Track	maximal aerobic	run to exhaustion	Comp.	6 (0 F, 6 M)	Mean: 21.6; SD: 4	Res., Freq., Spectral Energy, Sym. or Reg.
[104]	Leduc, et al.	2020	Outdoor	NR	5.00 m/s	240 m	Comp.	17 (0 F, 17 M)	Mean: 21.0; SD: 1.3	VT, AP, ML, Res., PlayerLoad
[105]	Lee, et al.	2010	Indoor	Treadmill	self-selected, 0.28 m/s above and below	15 min	Comp.	10 (4 F, 6 M)	Mean: 30; SD: 8	VT
[106]	Lee, et al.	2015	Indoor	Treadmill	2.0 and 3.5 m/s	30 s	Non	15 (0 F, 15 M)	Mean: 26.9; SD: 3.1	VT, AP, ML, Res., Seg. Rot. Velocity
[107]	Lin, et al.	2014	Indoor	Treadmill	1.67, 2.22, 2.50, and 3.33 m/s	4 min	Comp.	10 (0 F, 10 M)	Mean: 50.30; SD: 9.40	VT, AP, ML
			Outdoor	Not Controlled	not controlled	12 hours				
[108]	Lindsay, et al.	2016	Indoor	Treadmill	2.22, 2.78, and 3.33 m/s	8.5 min	Non	15 (4 F, 11 M)	Mean: 23.7; SD: 4.7	VT, AP, ML, Res.
[109]	Lindsay, et al.	2014	Indoor	Treadmill	2.22, 2.78, and 3.33 m/s	180 s	Non	18 (0 F, 18 M)	Mean: 24.0; SD: 4.2	VT, AP, ML, Res.
[110]	Lucas-Cuevas, et al.	2017	Indoor	Treadmill	2.22, 2.78, and 3.33 m/s	6 min	Rec.	30 (0 F, 30 M)	Mean: 27.3; SD: 6.4	VT, Shock-time, Shock-frequency, Freq., Spectral Energy
[111]	Lucas-Cuevas, et al.	2016	Indoor	Treadmill	60% maximal aerobic	3 min	Rec.	22 (NS)	Mean: 28.4; SD: 5.8	VT, Loading Rate, Shock-time
[112]	Lucas-Cuevas, et al.	2017	Indoor	Treadmill	3.33 m/s	16 min	Rec.	38 (18 F, 20 M)	Mean: 29.8; SD: 5.3	VT, Loading Rate, Shock-time
[113]	Macadam, et al.	2019	Indoor	Track	maximal sprint	160 m	Comp.	1 (0 F, 1 M)	Exact: 32; NA	Seg. Rot., Seg. Rot. Velocity
[114]	Macadam, et al.	2020	NR	NR	sprint	200 m	Comp.	15 (0 F, 15 M)	Mean: 21.0; SD: 2.5	Seg. Rot., Seg. Rot. Velocity
[115]	Macadam, et al.	2020	Indoor	Treadmill	maximal sprint	80 s	Non	14 (NS)	Mean: 24.9; SD: 4.2	Seg. Rot., Seg. Rot. Velocity
[116]	Macdermid, et al.	2017	Indoor	Treadmill	2.61 m/s	9 min	Rec.	8 (NS)	Mean: 25; SD: 12	Res., Loading Rate
[117]	Mangubat, et al.	2018	Indoor	Treadmill	preferred	90 s	Rec.	13 (5 F, 8 M)	Mean: 27.1; SD: 5.1	VT
[118]	Masci, et al.	2013	Indoor	Floor	maximal	15 m	Non	54 (NS)	Mean: 5; SD: 3	VT, AP, ML, Res., Stiffness
[119]	McGregor, et al.	2009	Indoor	Treadmill	incremental from 0.56 m/s	run to exhaustion	Comp.	7 (0 F, 7 M)	Mean: 21.4; SD: 1.7; Range: 19–24	VT, AP, ML, Res., Entropy
[120]	Mercer and Chona	2015	Indoor	Treadmill	100%, 110%, 120%, and 130% of preferred	16 min	Non	10 (6 F, 4 M)	Mean: 21.4; SD: 2.0	VT
[121]	Mercer, et al.	2002	Indoor	Treadmill	50%, 60%, 70%, 80%, 90%, and 100% of maximal	120 s	Non	8 (0 F, 8 M)	Mean: 25; SD: 4.6	VT, Shock-frequency
[122]	Mercer, et al.	2003	Indoor	Treadmill	3.8 m/s	140 s	Rec.	10 (0 F, 10 M)	Mean: 24; SD: 5.8	Shock-frequency, Spectral Energy
[123]	Mercer, et al.	2010	Indoor	Floor	preferred	200 m	Non	18 (7 F, 11 M)	Mean: 10.3; SD: 1	VT, Shock-time
			Indoor	Treadmill	preferred, 0.5 m/s faster and slower	6 min				
[124]	Mercer, et al.	2003	Indoor	Treadmill	3.1 and 3.8 m/s	40 s	Non	10 (0 F, 10 M)	Mean: 24; SD: 6	VT, Shock-frequency, Spectral Energy
[125]	Meyer, et al.	2015	Indoor	Floor	1.67 and 2.78 m/s	140 m	Non	13 (3 F, 10 M)	Mean: 10.1; SD: 3.0; Range: 5–16	VT
[126]	Meyer, et al.	2018	Indoor	Track	3.5 m/s	900 m	Non	36 (18 F, 18 M)	Mean: 25.4; SD: 3.5	VT, AP, ML, Variability-any axis
			Indoor	Treadmill	3.0 m/s	1 min				
			Outdoor	Grass	3.0 m/s	1 min				
			Outdoor	Pavement	3.0 m/s	1 min				
[127]	Mitschke, et al.	2017	Indoor	Track	2.5 and 3.5 m/s	450 m	Rec.	24 (NS)	Mean: 24.7; SD: 4.1	Seg. Rot. Velocity
[128]	Mitschke, et al.	2017	Indoor	Track	3.5 m/s	450 m	Rec.	21 (0 F, 21 M)	Mean: 28.9; SD: 10.8	VT, Seg. Rot., Seg. Rot. Velocity
[129]	Mitschke, et al.	2018	Indoor	Track	self-selected	225 m	Rec.	21 (0 F, 21 M)	Mean: 24.4; SD: 4.2	VT, Seg. Rot.
[130]	Montgomery, et al.	2016	Indoor	Floor	natural jog, natural run	40 m	Non	15 (NS)	Mean: 24.2; SD: 3.8	VT, Loading Rate
			Indoor	Treadmill	natural jog, natural run	1 min				
[131]	Moran, et al.	2017	Indoor	Treadmill	75% of maximum HR	37 min	Rec.	15 (6 F, 9 M)	Mean: 20.4; SD: 2.4	VT
[132]	Morrow, et al.	2014	Indoor	Floor	slow, normal, fast	≥504 m	Non	11 (8 F, 3 M)	Mean: 33.4; SD: 10.5	VT, AP, ML
[133]	Neugebauer, et al.	2012	Indoor	Floor	consistent	540 m	Non	35 (20 F, 15 M)	Mean: 11.6; SD: 0.7	Res.
[134]	Nüesch, et al.	2017	Indoor	Treadmill	self-selected	6 min	Rec.	20 (12 F, 8 M)	Mean: 27.4; SD: 8.3	Joint ROM
[135]	O’Connor and Hamill	2002	Indoor	Treadmill	3.8 m/s	4 min	Non	12 (0 F, 12 M)	Mean: 27.6; SD: 5.3	VT
[136]	Provot, et al.	2017	Indoor	Treadmill	2.22, 2.78, 3.33, 3.89, 4.44, and 5.00 m/s	7 min	Non	1 (0 F, 1 M)	Exact: 33; NA	Res., Spectral Energy
[137]	Provot, et al.	2019	Indoor	Treadmill	2.22, 2.50, 2.78, 3.06, 3.33, 3.61, 3.89, 4.44, and 5.00 m/s	9 min	Rec.	18 (8 F, 10 M)	Mean: 31.4; SD: 8.9	Res., Freq., Spectral Energy, Stiffness
[138]	Rabuffetti, et al.	2019	Indoor	Treadmill	1.8 and 2.2 m/s	140 s	Non	25 (11 F, 14 M)	Mean: 26.5; SD: 4.5; Range: 20–40	Sym. or Reg.
[139]	Raper, et al.	2018	Indoor	Track	slow, medium, fast	50 m	Comp.	10 (6 F, 4 M)	Mean: 26.90; SD: 4.03	VT
[140]	Reenalda, et al.	2019	Outdoor	Track	10 km pace	20 min	Rec.	10 (0 F, 10 M)	Mean: 31; SD: 5	VT, Shock-time, Joint ROM
[141]	Schütte, et al.	2015	Indoor	Treadmill	3.2 km pace	run to exhaustion	Rec.	20 (8 F, 12 M)	Mean: 21.05; SD: 2.14	VT, AP, ML, Axis Ratio, Sym. or Reg., Entropy
[142]	Schütte, et al.	2016	Outdoor	Pavement	self-selected	180 m	Rec.	28 (14 F, 14 M)	Mean: 22.62; SD: 3.07	VT, AP, Freq., Axis Ratio, Sym. or Reg., Entropy
			Outdoor	Track	self-selected	180 m				
			Outdoor	Trail	self-selected	180 m				
[143]	Schütte, et al.	2018	Indoor	Treadmill	incremental from 2.22 or 2.50 m/s	run to exhaustion	Rec.	30 (14 F, 16 M)	Mean: 21.75; SD: 1.40	VT, AP, ML, Axis Ratio, Sym. or Reg., Entropy
[144]	Setuain, et al.	2017	Indoor	Floor	maximal sprint	60 m	Comp.	1 (0 F, 1 M)	Exact: 19; NA	VT, AP, Res., COM Disp., COM Δv
[145]	Setuain, et al.	2018	Indoor	Floor	maximal sprint	80 m	Rec.	16 (8 F, 8 M)	Mean: 28.8; SD: 5.35	AP, Res., COM Δv, Power
[146]	Sheerin, et al.	2018	Indoor	Treadmill	2.7, 3.0, 3.3, and 3.7 m/s	8 min	Rec.	14 (0 F, 14 M)	Mean: 33.6; SD: 11.6	Res.
[147]	Sheerin, et al.	2018	Indoor	Treadmill	2.7, 3.0, 3.3, and 3.7 m/s	8 min	Rec.	85 (20 F, 65 M)	Mean: 39.51; SD: 8.92	Res.
[148]	Shiang, et al.	2016	Indoor	Treadmill	1.94, 2.78, and 3.61 m/s	6 min	Non	6 (0 F, 6 M)	Mean: 25.4; SD: 1.7	Seg. Rot.
[149]	Simoni, et al.	2020	Indoor	Treadmill	self-selected	1 min	Rec.	87 (28 F, 59 M)	Mean: 41; SD: 10	VT, AP, ML
[150]	Stickford, et al.	2015	Indoor	Treadmill	3.88, 4.47, and 5.00 m/s	12 min	Comp.	16 (0 F, 16 M)	Mean: 22.4; SD: 3	Stiffness, COM Disp.
[151]	TenBroek, et al.	2014	Indoor	Treadmill	3 m/s	90 min	Rec.	10 (0 F, 10 M)	NR; Range: 18–55	VT, Shock-frequency
[152]	Tenforde, et al.	2020	Indoor	Treadmill	self-selected	3 min	Rec.	169 (74 F, 95 M)	Mean: 38.7; SD: 13.1	VT, Res.
[153]	Thomas and Derrick	2003	Indoor	Treadmill	preferred	10 min	Comp.	12 (6 F, 6 M)	Mean: 20.9; SD: 2.3	VT, Shock-time, Spectral Energy, Joint ROM, Joint ω
[154]	Tirosh, et al.	2019	Indoor	Treadmill	20% above walking	27 min	Non	37 (NS)	Mean: 9.2; SD: 1.3	VT
[155]	Tirosh, et al.	2017	Indoor	Treadmill	20% above walking	2 min	Non	24 (NS)	Mean: 8.5; SD: 0.9	VT
[156]	Tirosh, et al.	2020	Indoor	Treadmill	20% above walking	2 min	Non	32 (15 F, 17 M)	Mean: 9.26; SD: 1.18	VT, Shock-time
[157]	van Werkhoven, et al.	2019	Indoor	Treadmill	comfortable	3 min	Non	12 (NS)	NR; Range: 18–45	Joint ROM, Joint ω, Seg. Rot.
[158]	Walsh	2021	Indoor	Treadmill	100% and 120% of preferred	12 min	Non	18 (9 F, 9 M)	Mean: 29; SD: 6.5	Stability, Entropy
[159]	Waite, et al.	2021	Outdoor	Grass	80% of 1 mile pace	180 m	Rec.	13 (5 F, 8 M)	Mean: 20.07; SD: 0.95	VT
			Outdoor	Pavement	80% of 1 mile pace	360 m				
[160]	Winter, et al.	2016	Indoor	Floor	self-selected	8.2 km	Rec.	10 (4 F, 6 M)	Mean: 27.5; SD: 9.5	VT, AP, ML
[161]	Wixted, et al.	2010	Outdoor	Track	race effort	1500 m	Comp.	2 (NS)	NR	VT, AP, ML
[162]	Wood and Kipp	2014	Indoor	Treadmill	comfortable fast jog	25 min	Rec.	9 (6 F, 3 M)	Mean: 20; SD: 1.5	VT, AP, Res.
[163]	Wundersitz, et al.	2015	Indoor	Treadmill	3.3, 5.0, and 5.9 m/s	90 s	Non	39 (11 F, 28 M)	Mean: 24.2; SD: 2.5	Res.
[164]	Zhang, et al.	2019	Indoor	Treadmill	90%, 100%, and 110% of preferred	9 min	Rec.	13 (3 F, 10 M)	Mean: 41.1; SD: 6.9	VT
[165]	Zhang, et al.	2016	Indoor	Treadmill	preferred, 15% faster and slower	18 min	Non	10 (2 F, 8 M)	Mean: 23.6; SD: 3.8	VT

Note: NR = not reported; Pavement = pavement or sidewalk; Floor = floor or platform; Rec. = recreational; Comp. = competitive; Non = non-runners; Disp. = displacement; Δv = change in velocity; Sym. or Reg. = symmetry or regularity; Res. = resultant magnitude; VT = vertical/axial magnitude; AP = anterior–posterior magnitude; ML = medial–lateral magnitude; Seg. Rot. = segment rotation; Shock—time = shock attenuation—time domain; Shock—frequency = shock attenuation—frequency domain; Joint ROM = joint angles or range of motion; Joint ω = joint angular velocity; Freq. = frequency content.

**Table 3 sensors-22-01722-t003:** Study characteristics where the analyzed distance is 200–1000 m.

Ref.	Author	Year	Location	Surface	Speed	Distance/Duration	Type	Number (Sex)	Overall Age	Metric(s)
[166]	Aubry, et al.	2018	Indoor	Treadmill	3.06–5.00 m/s	6 min	Comp.	11 (0 F, 11 M)	Mean: 33.4; SD: 6.6	COM Disp., Power
			Outdoor	Track	3.06–5.00 m/s	12 min	Rec.	13 (0 F, 13 M)		
[167]	Austin, et al.	2018	Indoor	Treadmill	85–89% VO2max	8 min	Comp.	17 (8 F, 9 M)	Mean: 20.6; SD: 2.3	Power
[168]	Barrett, et al.	2014	Indoor	Treadmill	incremental from 1.94 m/s	2 runs to exhaustion	Comp.	44 (NS)	Mean: 22; SD: 3	PlayerLoad
[5]	Benson, et al.	2020	Indoor	Treadmill	preferred	5 min	Rec.	69 (31 F, 38 M)	Mean: 33.7; SD: 11.5	VT, AP, ML, Res., Sym. or Reg.
			Outdoor	Pavement	preferred	600 m				
[169]	De Brabandere, et al.	2018	Indoor	Treadmill	incremental from 2.5 m/s to 4.58 m/s	run to exhaustion	Rec.	28 (16 F, 12 M)	Mean: 21.8; SD: 1.3	VT, AP, ML
[170]	Cher, et al.	2017	Indoor	Treadmill	slow, medium, fast	60 min	Non	12 (4 F, 8 M)	Mean: 29.4; SD: 6.8	VT, AP, ML, Res.
[171]	Clansey, et al.	2016	Indoor	Treadmill	95% of onset of blood lactate accumulation	40 min	Rec.	13 (0 F, 13 M)	Mean: 35.1; SD: 10.2	VT, Shock-frequency
[172]	Clermont, et al.	2019	Indoor	Treadmill	preferred	5 min	Rec.	41 (16 F, 25 M)	Mean: 32.5; SD: 12.7	VT, AP, ML, Res., Sym. or Reg.
[173]	Clermont, et al.	2020	Indoor	Track	self-selected	21 min	Rec.	17 (10 F, 7 M)	Mean: 39.8; SD: 9.6	VT, AP, ML, Res., Sym. or Reg.
			Outdoor	Track	not controlled	6 km				
[174]	Deriaz, et al.	2010	Indoor	Treadmill	comfortable	15 min	Rec.	65 (0 F, 65 M)	Mean: 36.0; SD: 6.8	VT
							Non	16 (0 F, 16 M)		
[175]	Enders, et al.	2014	Indoor	Treadmill	3.5 m/s	20 min	Rec.	12 (0 F, 12 M)	Mean: 25.56; SD: 2.88	VT, Shock-frequency, Freq., Spectral Energy
[176]	Garcia-Byrne, et al.	2020	Indoor	Floor	2.22 m/s	400 m	Comp.	36 (0 F, 36 M)	Mean: 25; SD: 3	PlayerLoad
[177]	Giandolini, et al.	2016	Outdoor	Pavement	self-selected	6.5 km	Rec.	23 (0 F, 23 M)	Mean: 39; SD: 11	VT, ML, Res., Shock-time, Shock-frequency
			Outdoor	Trail	self-selected	6.5 km				
[178]	Giandolini, et al.	2017	Outdoor	Pavement	self-selected	6.5 km	Rec.	23 (0 F, 23 M)	Mean: 39; SD: 11	VT
			Outdoor	Trail	self-selected	6.5 km				
[179]	Horvais, et al.	2019	Indoor	Treadmill	3.9 m/s	14 min	Rec.	10 (0 F, 10 M)	Mean: 27.3; SD: 5.4	VT, AP, Res., Spectral Energy
[180]	Hughes, et al.	2019	Indoor	Treadmill	3.89 and 5.00 m/s	9 min	Comp.	16 (NS)	Mean: 17.36; SD: 1.25	VT
[181]	Koska, et al.	2018	Indoor	Treadmill	2.78, 3.33, and 4.17 m/s	9 min	Rec.	51 (15 F, 36 M)	Mean: 33.9; SD: 8.2	Seg. Rot., Seg. Rot. Velocity
[182]	Melo, et al.	2020	Indoor	Treadmill	10 km pace	20 km	Rec.	13 (5 F, 8 M)	Mean: 36; SD: 4	COM Disp.
[183]	Moltó, et al.	2020	Indoor	Treadmill	typical	3 min	Rec.	38 (16 F, 22 M)	Mean: 26.7; SD: 7.7	Seg. Rot.
[184]	Morio, et al.	2016	Indoor	Treadmill	3.06 m/s	12 min	Rec.	8 (0 F, 8 M)	Mean: 26; SD: 2	VT, AP, ML
[185]	Murray, et al.	2017	Indoor	Treadmill	incremental to reach [La]_b_ concentration of 4 mmol/L	run to exhaustion	Comp.	6 (0 F, 6 M)	Mean: 15.6; SD: 1.2	VT, AP, ML, Entropy
[186]	Navalta, et al.	2019	Outdoor	Trail	self-selected	10 min	Non	20 (8 F, 12 M)	Mean: 22.2; SD: 5.8	Stiffness, COM Disp., Power
[187]	Olin and Gutierrez	2013	Indoor	Treadmill	comfortable	21 min	Rec.	18 (12 F, 6 M)	Mean: 31.2; SD: 7.9	VT
[188]	Perrotin, et al.	2021	Outdoor	Pavement	not controlled	1 km	Rec.	30 (3 F, 27 M)	Mean: 36.4; SD: 8	Stiffness, COM Disp., Power
[189]	Provot, et al.	2016	Indoor	Treadmill	3.33 m/s	40 min	Rec.	1 (0 F, 1 M)	Exact: 22; NA	Res., Freq., Spectral Energy, Stiffness
[190]	Reenalda, et al.	2016	Outdoor	Pavement	not controlled	42.2 km	Rec.	3 (0 F, 3 M)	Mean: 38.7; SD: 8.2; Range: 31–50	VT, Joint ROM, COM Disp.
[191]	Seeley, et al.	2020	Indoor	Treadmill	2.68, 3.13, and 3.58 m/s	12 min	Non	31 (14 F, 17 M)	Mean: 23; SD: 3	VT, AP, ML, Res.
[192]	Shih, et al.	2014	Indoor	Treadmill	70% of maximal	30 min	Rec.	15 (NS)	Mean: 24.5; SD: 1.7	Joint ROM, Joint ω
[193]	Tirosh, et al.	2019	Indoor	Floor	20% above walking	1 km	Non	10 (NS)	Mean: 10.7; SD: 1.27	VT
			Outdoor	Grass	3.33, 3.89, 4.44, and 5.00 m/s	12 min				
			Outdoor	Pavement	3.33, 3.89, 4.44, and 5.00 m/s	12 min				
			Outdoor	Track	3.33, 3.89, 4.44, and 5.00 m/s	12 min				
			Outdoor	Track	4.78 m/s	6 km				
[194]	Ueberschar, et al.	2019	Indoor	Treadmill	incremental from 1.67 m/s	run to exhaustion	Rec.	15 (0 F, 15 M)	Mean: 30; SD: 7	VT, Res.
[195]	Van den Berghe, et al.	2021	Indoor	Track	3.2 m/s	20 min	Rec.	10 (NS)	Mean: 33; SD: 9; Range: 24–49	VT
[196]	van der Bie and Krose	2015	Indoor	Treadmill	ventilatory threshold	run to exhaustion	Non	18 (14 F, 4 M)	Mean: 23; SD: 3	VT, AP, ML, Variability-any axis, Entropy
[9]	Watari, et al.	2016	Indoor	Treadmill	2.7, 3.0, 3.3, 3.6, and 3.9 m/s	5 min	Rec.	22 (8 F, 14 M)	Mean: 28.2; SD: 10.1	COM Disp.
[197]	Weich, et al.	2019	Indoor	Track	95% anaerobic threshold	5000 m	Rec.	25 (NS)	Mean: 26.64; SD: 6.86	VT, AP, ML
							Non	9 (NS)		

Note: NR = not reported; Pavement = pavement or sidewalk; Floor = floor or platform; Rec. = recreational; Comp. = competitive; Non = non-runners; Disp. = displacement; Δv = change in velocity; Sym. or Reg. = symmetry or regularity; Res. = resultant magnitude; VT = vertical/axial magnitude; AP = anterior–posterior magnitude; ML = medial–lateral magnitude; Seg. Rot. = segment rotation; Shock—time = shock attenuation—time domain; Shock—frequency = shock attenuation—frequency domain; Joint ROM = joint angles or range of motion; Joint ω = joint angular velocity; Freq. = frequency content.

**Table 4 sensors-22-01722-t004:** Study characteristics where the analyzed distance is >1000 m over a single run.

Ref.	Author	Year	Location	Surface	Speed	Distance/Duration	Type	Number (Sex)	Overall Age	Metric(s)
[198]	Bigelow, et al.	2013	Indoor	Track	self-selected	4 miles	Rec.	12 (NS)	Mean: 32.8; SD: 9.8	VT, ML
			Indoor	Treadmill	self-selected	4 miles				
[199]	Brahms, et al.	2020	Indoor	Track	5 km pace	run to exhaustion	Comp.	16 (NS)	Mean: 24; SD: 3.9	Res.
							Rec.	16 (NS)		
			Indoor	Treadmill	incremental from 2.5 m/s	run to exhaustion				
			Outdoor	Track	2.78 m/s	9 min				
			Outdoor	Track	incremental from 2.5 m/s	run to exhaustion				
[200]	Clermont, et al.	2019	Outdoor	Not Controlled	not controlled	42.2 km	Rec.	27 (15 F, 12 M)	Mean: 45.1; SD: 11.5	Joint ROM, Seg. Rot., COM Disp., COM Δv
[201]	DeJong and Hertel	2020	Outdoor	Pavement	not controlled	6–21.1 km	Comp.	5 (4 F, 1 M)	Mean: 30.2; SD: 3.3	VT, AP, Joint ω
			Outdoor	Trail	not controlled	5 km				
[202]	Giandolini, et al.	2015	Outdoor	Trail	not controlled	45 km	Comp.	1 (0 F, 1 M)	Exact: 26; NA	VT, AP, Res., Freq.
[203]	Gómez-Carmona, et al.	2020	Indoor	Treadmill	incremental from 2.22 m/s	run to exhaustion	Rec.	20 (0 F, 20 M)	Mean: 27.32; SD: 6.65	PlayerLoad
			Outdoor	Track	incremental from 2.22 m/s	run to exhaustion				
[204]	Hoenig, et al.	2019	NR	Track	maximal	5000 m	Rec.	30 (0 F, 30 M)	Mean: 27; SD: 6.0	Stability
[205]	Provot, et al.	2019	Indoor	Treadmill	3.75 m/s	run to exhaustion	Rec.	10 (5 F, 5 M)	Mean: 38.0; SD: 11.6	VT, AP, ML, Res., Freq., Spectral Energy, Stiffness
[206]	Rojas-Valverde, et al.	2019	Outdoor	Trail	not controlled	35.27 km	Rec.	20 (0 F, 20 M)	Mean: 38.95; SD: 9.99	Res., Entropy, PlayerLoad
[207]	Rojas-Valverde, et al.	2020	Outdoor	Not Controlled	not controlled	36 km	Rec.	18 (NS)	Mean: 38.78; SD: 10.38	Res.
[208]	Schütte, et al.	2018	Outdoor	Track	3.2 km pace	3200 m	Rec.	16 (6 F, 10 M)	Mean: 20.23; SD: 0.78	VT, Shock-frequency, Spectral Energy, Axis Ratio, Sym. or Reg., Entropy
							Rec.	14 (6 F, 8 M)		
[209]	Ueberschar, et al.	2019	Outdoor	Trail	3.17 m/s	10 km	Non	10 (NS)	Mean: 10.7; SD: 1.27	VT

Note: NR = not reported; Pavement = pavement or sidewalk; Floor = floor or platform; Rec. = recreational; Comp. = competitive; Non = non-runners; Disp. = displacement; Δv = change in velocity; Sym. or Reg. = symmetry or regularity; Res. = resultant magnitude; VT = vertical/axial magnitude; AP = anterior–posterior magnitude; ML = medial–lateral magnitude; Seg. Rot. = segment rotation; Shock—time = shock attenuation—time domain; Shock—frequency = shock attenuation—frequency domain; Joint ROM = joint angles or range of motion; Joint ω = joint angular velocity; Freq. = frequency content.

**Table 5 sensors-22-01722-t005:** Study characteristics where the analyzed distance is >1000 m over multiple runs.

Ref.	Author	Year	Location	Surface	Speed	Distance/Duration	Type	Number (Sex)	Overall Age	Metric(s)
[210]	Ahamed, et al.	2019	Outdoor	Not Controlled	not controlled	29 km	Rec.	11 (10 F, 1 M)	Mean: 44.1; SD: 9.1	Joint ROM, Seg. Rot., COM Disp., COM Δv
[211]	Ahamed, et al.	2018	Outdoor	Pavement	not controlled	≥24 km	Rec.	6 (5 F, 1 M)	Mean: 44.4; NR	Seg. Rot., COM Disp., COM Δv
[212]	Ahamed, et al.	2019	Outdoor	Not Controlled	not controlled	7 runs	Rec.	35 (25 F, 10 M)	Mean: 49.7; SD: 9.6	Joint ROM, Seg. Rot., COM Disp., COM Δv
[213]	Benson, et al.	2019	Outdoor	Not Controlled	not controlled	10 runs	Rec.	12 (9 F, 3 M)	Mean: 48.5; SD: 12.0	Joint ROM, Seg. Rot., COM Disp., COM Δv
[214]	Carton-Llorente, et al.	2021	Indoor	Treadmill	submaximal	120 min	Rec.	22 (0 F, 22 M)	Mean: 34; SD: 7.5	Power
[215]	Cerezuela-Espejo, et al.	2020	Indoor	Treadmill	2.78 m/s	24 min	Rec.	12 (0 F, 12 M)	Mean: 25.7; SD: 7.9	Power
[216]	Colapietro, et al.	2020	Outdoor	Track	slow, fast	3200 m	Rec.	18 (10 F, 8 M)	Mean: 22.7; SD: 4.7	VT, AP, Joint ROM, Joint ω
[217]	Gregory, et al.	2019	Outdoor	Track	hard	1600 m	Non	12 (6 F, 6 M)	Mean: 22.0; SD: 1.9	VT, AP, Joint ROM, Joint ω
[218]	Hollander, et al.	2021	Indoor	Treadmill	70% VO2max	105 min	Non	41 (20 F, 21 M)	Mean: 25.2; SD: 3.1	Stability
[219]	Hollis, et al.	2019	Outdoor	Grass	moderate, hard	3200 m	Rec.	15 (8 F, 7 M)	Mean: 20; SD: 3.1	VT, AP, Joint ROM, Joint ω
			Outdoor	Track	moderate, hard	3200 m				
[220]	Kiernan, et al.	2018	Outdoor	Not Controlled	not controlled	60 day training period	Comp.	9 (0 F, 9 M)	Mean: 18.7; SD: 1.0	VT
[221]	Koldenhoven, et al.	2020	Outdoor	Not Controlled	not controlled	3 runs	Rec.	16 (8 F, 8 M)	Mean: 23.5; SD: 5	AP, Res., Seg. Rot., Seg. Rot. Velocity
[116]	Macdermid, et al.	2017	Indoor	Treadmill	2.83 m/s	30 min	Comp.	6 (NS)	Mean: 29.8; SD: 13.0	Loading Rate, Shock-time, COM Disp.
[222]	McGregor, et al.	2009	Indoor	Treadmill	incremental from 0.56 m/s	2 runs to exhaustion	Comp.	7 (0 F, 7 M)	Mean: 26.5; SD: 5.7	VT, AP, ML, Res.
							Non	7 (NS)		
[223]	Nüesch, et al.	2019	Indoor	Treadmill	self-selected	15 min	Rec.	19 (11 F, 8 M)	Mean: 27.7; SD: 8.6	Joint ROM
[224]	Olcina, et al.	2019	NR	Track	maximal	24 min	Comp.	10 (2 F, 8 M)	Mean: 25.7; SD: 8.9	COM Disp.
			Outdoor	Trail	not controlled	4.1 km				
[225]	Rochat, et al.	2019	Outdoor	Trail	not controlled	15 km	Rec.	9 (0 F, 9 M)	Mean: 37.8; SD: 7	VT, COM Disp.
[226]	Ruder, et al.	2019	Outdoor	Pavement	not controlled	42.2 km	Rec.	222 (103 F, 119 M)	Mean: 44.1; SD: 10.8	VT
[227]	Ryan, et al.	2021	Outdoor	Not Controlled	not controlled	10–12 runs	Rec.	12 (0 F, 12 M)	NR; Range: 14–18	Res.
[228]	Strohrmann, et al.	2012	Indoor	Treadmill	85% of maximal	45 min	Rec.	21 (NS)	NR	Res., Joint ω, Seg. Rot., COM Disp.
			Outdoor	Track	85% of maximal	45 min				
[209]	Ueberschar, et al.	2019	Outdoor	Pavement	3.64 and 6.67 m/s	10.7 km	Comp.	2 (0 F, 2 M)	Mean: 17.6; SD: 1.13	VT
[229]	Van den Berghe, et al.	2020	Indoor	Track	3.2 m/s	24.5 min	Rec.	10 (5 F, 5 M)	Mean: 33; SD: 9	VT
[230]	Vanwanseele, et al.	2020	Indoor	Treadmill	2.22, 3.33, and 4.44 m/s	3 min	Rec.	68 (NS)	Mean: 29.5; SD: 8.1	VT, AP, ML
			Outdoor	Not Controlled	not controlled	3 month training period				
[231]	Willis, et al.	2019	Indoor	Treadmill	2.50, 2.78, 3.61, and 4.17 m/s	10 min	Comp.	12 (NS)	Mean: 33.6; SD: 4.3	Stiffness

Note: NR = not reported; Pavement = pavement or sidewalk; Floor = floor or platform; Rec. = recreational; Comp. = competitive; Non = non-runners; Disp. = displacement; Δv = change in velocity; Sym. or Reg. = symmetry or regularity; Res. = resultant magnitude; VT = vertical/axial magnitude; AP = anterior–posterior magnitude; ML = medial–lateral magnitude; Seg. Rot. = segment rotation; Shock—time = shock attenuation—time domain; Shock—frequency = shock attenuation—frequency domain; Joint ROM = joint angles or range of motion; Joint ω = joint angular velocity; Freq. = frequency content.

**Table 6 sensors-22-01722-t006:** Study characteristics where the analyzed distance is not reported.

Ref.	Author	Year	Location	Surface	Speed	Distance/Duration	Type	Number (Sex)	Overall Age	Metric(s)
[232]	Bielik	2019	Outdoor	Track	2.78, 3.33, and 4.17 m/s	15 min	Rec.	73 (24 F, 49 M)	Mean: 29.2; SD: 4.1	COM Disp.
			Outdoor	Treadmill	incremental from 0.28 m/s	run to exhaustion				
[233]	Bielik and Clementis	2017	Indoor	Treadmill	incremental from 2.22 m/s	run to exhaustion	Comp.	30 (NS)	NR	COM Disp.
							Rec.	24 (NS)		
[234]	Butler, et al.	2007	Indoor	Treadmill	self-selected	2 runs	Rec.	24 (NS)	Mean: 21.4; SD: 3.1	VT
[235]	Cooper, et al.	2009	Indoor	Treadmill	incremental from 0.45 m/s to 2.24 m/s	5 min	Non	7 (2 F, 5 M)	Mean: 30; SD: 6	Joint ROM
[236]	de Fontenay, et al.	2020	Indoor	Treadmill	self-selected	NR	Non	32 (13 F, 19 M)	Mean: 27.0; SD: 5.5	AP, Loading Rate, Joint ROM, COM Disp.
[237]	Dufek, et al.	2008	Indoor	Treadmill	self-selected	90 s	Rec.	31 (31 F, 0 M)	Mean: 26.7; SD: 3.8	VT, Shock-time
[238]	Garrett, et al.	2019	NR	NR	6.25 m/s	450 m	Comp.	23 (0 F, 23 M)	Mean: 22.4; SD: 3.6	PlayerLoad
[239]	Gurchiek, et al.	2017	Indoor	Floor	sprint	NR	Non	15 (3 F, 12 M)	Mean: 23.2; SD: 2.11	Seg. Rot.
[240]	Sheerin, et al.	2020	Indoor	Treadmill	comfortable	NR	Rec.	18 (7 F, 11 M)	Mean: 35.2; SD: 9.6	Res.
			Outdoor	Track	comfortable	800 m				
[241]	Ueberschar, et al.	2019	Indoor	Treadmill	incremental	run to exhaustion	Comp.	53 (18 F, 35 M)	Mean: 20.07; SD: 3.65	VT, Shock-time
[242]	Zadeh, et al.	2020	Outdoor	Not Controlled	not controlled	36 physical training sessions	Non	55 (16 F, 39 M)	Mean: 20.8; SD: 3.32	VT, Res., Loading Rate

Note: NR = not reported; Pavement = pavement or sidewalk; Floor = floor or platform; Rec. = recreational; Comp. = competitive; Non = non-runners; Disp. = displacement; Δv = change in velocity; Sym. or Reg. = symmetry or regularity; Res. = resultant magnitude; VT = vertical/axial magnitude; AP = anterior–posterior magnitude; ML = medial–lateral magnitude; Seg. Rot. = segment rotation; Shock—time = shock attenuation—time domain; Shock—frequency = shock attenuation—frequency domain; Joint ROM = joint angles or range of motion; Joint ω = joint angular velocity; Freq. = frequency content.

## Data Availability

All data are available within this manuscript.

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
