# Peer review of "Is This the Real Life, or Is This Just Laboratory? A Scoping Review of IMU-Based Running Gait Analysis"

_sensors, 2022, doi:10.3390/s22051722_

Round 1

Reviewer 1 Report

It is an interesting scoping review that investigates how IMUs are used to record running biomechanics. Since it is a scoping review the accuracy of the IMU measurements in the different studies were not questioned, which is a little short coming of the review. However, still an interesting review.

It has some interesting analyses and I have only a few comments.

Page 26 lines 283-285: a bit unclear that you first state triaxial accelerometer on the shank (18%) and later shank (35%) later in the next sentence. Suggest to rewrite this part.

Page 30 lines 381-388. Perhaps most studies are performed in the laboratory because the IMUs were tested against other sensors for reliability. The best way to test this is in a laboratory to avoid confounding variables. This is what I miss in the discussion as an explanation for the many studies in the laboratory. So add this to the discussion.

Author Response

Thank you for your comments and suggestions. We have addressed each of them below and indicated the changes in the manuscript with bold, underlined, and highlighted text.

It is an interesting scoping review that investigates how IMUs are used to record running biomechanics. Since it is a scoping review the accuracy of the IMU measurements in the different studies were not questioned, which is a little short coming of the review. However, still an interesting review.

Yes, we agree that a scoping review is not able to provide information that would result from a systematic review with meta-analysis. We have added this to the limitations (Line 429). We do think that a scoping review is appropriate considering the goal of this manuscript is to provide an overarching view of the field of running biomechanics and IMUs.

It has some interesting analyses and I have only a few comments.

Page 26 lines 283-285: a bit unclear that you first state triaxial accelerometer on the shank (18%) and later shank (35%) later in the next sentence. Suggest to rewrite this part.

Thank you for identifying this potentially confusing information. The combination of a triaxial accelerometer on the shank occurred 18% of the time. When looking at body parts regardless of the sensor or number of axes, the shank was used 35% of the time. This is because a uniaxial accelerometer is also commonly used on the shank (not just a triaxial accelerometer). We have clarified that the 35% refers to all sensors and all axes (Line 224).

Page 30 lines 381-388. Perhaps most studies are performed in the laboratory because the IMUs were tested against other sensors for reliability. The best way to test this is in a laboratory to avoid confounding variables. This is what I miss in the discussion as an explanation for the many studies in the laboratory. So add this to the discussion.

We agree that many of the included studies were conducted in indoor settings because the purpose of the study was to evaluate the validity and reliability of IMU-based metrics compared to metrics from force plates or motion capture systems. Please see the last paragraph in section 4.2 for further elaboration on this topic (Line 356-364).

Reviewer 2 Report

Review report – 1601463 – Sensors

A brief summary

The purpose of  the scoping review was to describe how IMUs are used to record running biomechanics in both laboratory and real-world conditions. Authors suggest that future studies should move out of the lab to less controlled and more real-world environments. By identifying the scope of IMU-based running biomechanics studies, authors aim to mark the progress made and the steps that remain for analysing running gait in real-world settings.

Broad comments

Introducing overview was supported by methodology used to fulfil aim of the study. Study design was appropriate, followed and supported by relevant and precise conclusions.  Limitations of the study were sufficiently explained. Very focused approach will take reader to improved insight of running biomechanics issues with regard to IMU-produced data.

Specific comments.

  • Minor spell check, and style corrections needed.
  • Ln 64-5:”review was designed to capture all journal articles that used IMUs to assess gait 64 quality during running, published in English since 2001. Exclusion criteria were…”-in one sentence, please describe what important indirect finding may have been sacrificed by this very focused approach (nicely limited). E.g. in certain circumstances, one space-oriented research may offer more important finding than scoped review on ‘micro-cosmic’ topic…
  • Ln153-6:” When the axis for tibial acceleration magnitude was not specified, it was assumed to be vertical. For other situations where the axis of acceleration magnitude was not reported it was assumed to be the resultant.” – it would be acceptable if criterion for this assumption was introduced, otherwise, it may mislead and blur general (very concise and impressive work done in general)
  • Ln295-8:” Most devices were research-grade. Devices that are commercially available and designed for public use include: adidas Run Genie, Catapult, DorsaVi, Garmin, Google Nexus, Lumo Run, Milestone Pod, Polar, RunScribe, Runteq Zoi, Stryd, and Zephyr Bio-Harness. These devices were commonly worn on the shoe or lower or upper back.” – If possible, please visually present distribution of research with regard to producer – in most unbiased manner
  • Merge two graphs within Figure 5 into one graph (By adding one column at the end or in the middle of the graph)
  • Ln384-8:” It is unclear why researchers are using IMUs to record running, but still have participants running in the laboratory, at controlled speeds, on treadmills and/or over short distances. If the purpose of these d vices is to capture real-world running, we suggest that the research in this area should move out of the lab to less controlled environments.” – It is not unclear since training program for recreational purposes is often anchored within the logic of the researcher. Contrary to competition surrounding recreational running is often practiced on a treadmill or likewise. If considered as laboratory session, training session becomes controllable within followed variables. Therefore, please rephrase!!

Author Response

Thank you for your comments and suggestions. We have addressed each of them below and indicated the changes in the manuscript with bold, underlined, and highlighted text.

A brief summary

The purpose of  the scoping review was to describe how IMUs are used to record running biomechanics in both laboratory and real-world conditions. Authors suggest that future studies should move out of the lab to less controlled and more real-world environments. By identifying the scope of IMU-based running biomechanics studies, authors aim to mark the progress made and the steps that remain for analysing running gait in real-world settings.

Broad comments

Introducing overview was supported by methodology used to fulfil aim of the study. Study design was appropriate, followed and supported by relevant and precise conclusions.  Limitations of the study were sufficiently explained. Very focused approach will take reader to improved insight of running biomechanics issues with regard to IMU-produced data.

Specific comments.

Minor spell check, and style corrections needed.

We have reviewed the document and made minor spelling changes.

Ln 64-5:”review was designed to capture all journal articles that used IMUs to assess gait 64 quality during running, published in English since 2001. Exclusion criteria were…”-in one sentence, please describe what important indirect finding may have been sacrificed by this very focused approach (nicely limited). E.g. in certain circumstances, one space-oriented research may offer more important finding than scoped review on ‘micro-cosmic’ topic…

Thank you for this suggestion. We believe we have outlined the limitations of the inclusion and exclusion criteria, specifically the focus on IMUs as the only type of wearable technology, excluding studies where only spatiotemporal gait characteristics (not gait quality) were reported, and excluding studies that focused on the development of new technologies or methods. Rather than state the limitations in the methods, we have expanded on them in the discussion. Please see section 4.5 (Line 416-428).

Ln153-6:” When the axis for tibial acceleration magnitude was not specified, it was assumed to be vertical. For other situations where the axis of acceleration magnitude was not reported it was assumed to be the resultant.” – it would be acceptable if criterion for this assumption was introduced, otherwise, it may mislead and blur general (very concise and impressive work done in general)

Thank you for this suggestion. We have clarified this criterion due to the large number of studies investigating shock absorption using an accelerometer placed on the tibia (Line 153-155).

Ln295-8:” Most devices were research-grade. Devices that are commercially available and designed for public use include: adidas Run Genie, Catapult, DorsaVi, Garmin, Google Nexus, Lumo Run, Milestone Pod, Polar, RunScribe, Runteq Zoi, Stryd, and Zephyr Bio-Harness. These devices were commonly worn on the shoe or lower or upper back.” – If possible, please visually present distribution of research with regard to producer – in most unbiased manner

Thank you for this suggestion. The frequency of any one producer of commercially available devices is small. We did, however, indicate the percentage of devices that were research-grade vs. consumer-focused (Line 229-231).

Merge two graphs within Figure 5 into one graph (By adding one column at the end or in the middle of the graph)

We apologize for the duplication of figures in Figure 5. When the folder of figures was zipped to be attached to the manuscript submission, a previous figure version was also included. We have removed the duplicate figure.

Ln384-8:” It is unclear why researchers are using IMUs to record running, but still have participants running in the laboratory, at controlled speeds, on treadmills and/or over short distances. If the purpose of these d vices is to capture real-world running, we suggest that the research in this area should move out of the lab to less controlled environments.” – It is not unclear since training program for recreational purposes is often anchored within the logic of the researcher. Contrary to competition surrounding recreational running is often practiced on a treadmill or likewise. If considered as laboratory session, training session becomes controllable within followed variables. Therefore, please rephrase!!

We agree that some runners train on a treadmill and therefore treadmill running may be considered a real-world setting. However, the treadmill running in the included studies was not an individual’s typical training setting. The treadmill running occurred in a laboratory, under the supervision and direction of research personnel, and often included prescribed speeds and distances. Therefore, it is difficult to say that any laboratory-based running is representative of a typical training session. We argue that the best way to capture a typical training session is to provide a runner with an IMU and have them wear that whenever and however they run, which may include treadmill sessions at their own pace, distance and motivation.